# Usability and acceptability of a two-way texting intervention for post-operative follow-up for voluntary medical male circumcision in Zimbabwe

**Caryl Feldacker**[1,2]*, **Isaac Holeman**[1,3], **Vernon Murenje**[4], **Sinokuthemba Xaba**[5], **Michael Korir**[3], **Bill Wambua**[3], **Batsirai Makunike-Chikwinya**[4], **Marianne Holec**[2], **Scott Barnhart**[1,2,6], **Mufuta Tshimanga**[7]

1 Department of Global Health, University of Washington, Seattle, WA, United States of America,
2 International Training and Education Center for Health (I-TECH), Seattle, WA, United States of America,
3 Medic Mobile, Nairobi, Kenya, 4 International Training and Education Center for Health (I-TECH), Harare, Zimbabwe, 5 Ministry of Health and Child Care, Harare, Zimbabwe, 6 Department of Medicine, University of Washington, Seattle, WA, United States of America, 7 Zimbabwe Community Health Intervention Project (ZiCHIRE), Harare, Zimbabwe

* cfeld@uw.edu

**Data Availability Statement:** All relevant quantitative data from the client usability surveys are within the manuscript and its Supporting

## Abstract

### Background

Voluntary medical male circumcision (MC) is safe and effective. Nevertheless, MC programs require multiple post-operative visits. In Zimbabwe, a randomized control trial (RCT) found that post-operative two-way texting (2wT) between clients and MC providers instead of in-person reviews reduced provider workload and safeguarded patient safety. A critical component of the RCT assessed usability and acceptability of 2wT among providers and clients. These findings inform scale-up of the 2wT approach to post-operative follow-up.

### Methods

The RCT assigned 362 adult MC clients with cell phones into 2wT; these men responded to 13 automated daily texts supported by interactive texting or in-person follow-up, when needed. A subset of 100 texting clients filled a self-administered usability survey on day 14. 2wT acceptability was ascertained via 2wT response rates. Among 2wT providers, eight key informant interviews focused on 2wT acceptability and usability. Influences of wage and age on response rates and client-reported potential AEs were explored using linear and logistic regression models, respectively.

### Results

Clients felt confident, comfortable, satisfied, and well-supported with 2wT-based follow-up; few noted texting challenges or concerns about healing. Clients felt 2wT saved them time and money. Response rates (92%) suggested 2wT acceptability. Both clients and providers felt 2wT was highly usable. Providers noted 2wT saved them time, empowered clients to

Information files. Our complete transcripts contain data that is sensitive or includes identifying information. We also believe that the Zimbabwe MoHCC would like the confidentiality of the participants protected in accordance with the consent agreement. Due to these concerns, we are unable to make the full transcripts available to a wider audience. We will make the transcripts easily available to fellow researchers or reviewers who complete a data sharing agreement. The edited transcripts will be available on a case by case basis after reviewing all materials for any potentially identifying details. Interested researchers may contact the corresponding author, CF, at cfeld@uw.edu for copies of the transcripts. Or, researchers could also contact Jane Edelson jedelson@uw.edu, Regulatory Specialist at the UW, for access to the transcripts.

**Funding:** Research reported in this publication was supported by the Fogarty International Center of the National Institutes of Health under Award Number R21TW010583 (CF). The content is solely the responsibility of the authors and does not necessarily represent the official views of the National Institutes of Health. The funders had no role in study design, data collection and analysis, decision to publish, or preparation of the manuscript.

**Competing interests:** The authors have declared that no competing interests exist.

engage in their healing, and closed gaps in MC service quality. For scale, providers reinforced good post-operative counseling on AEs and texting instructions. Wage and age did not influence text response rates or potential AE texts.

## Conclusion

Results strongly suggest that 2wT is highly usable and acceptable for providers and patients. Men with concerns solicited provider guidance and reassurance offered via text. Providers noted that men engaged proactively in their healing. 2wT between providers and patients should be expanded for MC and considered for other short-term care contexts.

The trial is registered on ClinicalTrials.gov, trial NCT03119337, and was activated on April 18, 2017. https://clinicaltrials.gov/ct2/show/NCT03119337

## Introduction

Voluntary medical male circumcision (MC) is a critical HIV prevention intervention with global support for expansion across sub-Saharan Africa (SSA) [1–8]. Over 22 million MCs were performed across 14 sub-Saharan African (SSA) countries between 2008–2018 [9]. MC is cost-effective for HIV prevention [10–13] making investment in MC a priority across SSA [12, 14–17]. However, severe health system constraints threaten the quality and pace of MC scale-up [18–20], reducing the likelihood of meeting the ambitious global target of 5 million annual MCs in SSA [21]. Failure to reach MC targets means fewer infections averted [22, 23]. In addition, PEPFAR and UNAIDS identify scarcity of resources as a major risk factor threatening further control of the HIV epidemic [24]. New, cost-effective innovations that improve efficiency while maintaining or improving MC quality are crucial to attain HIV prevention milestones and safeguard scarce HIV epidemic control resources [25].

MC is safe and major complications are rare; thus, most follow-up visits are unnecessary [26, 27]. Adverse events (AEs) from gold-standard active surveillance [25, 28, 29] report 5–7% AEs [4, 30–32] while routine passive surveillance reports below the commonly-accepted 2% AE rate [33–37]. To ensure client safety, current global standards for high quality MC services nevertheless require at least one follow-up within 14 days [38, 39]. Post-operative guidelines in many countries, including those in Zimbabwe[40], strongly suggest three post-operative visits. However, with low AE rates, overstretched clinic staff likely waste invaluable resources conducting reviews for MC clients without complications. For clients, men healing well needlessly pay for transport, miss work, and wait for reviews. Innovations that safely target MC follow-up only for men with potential AEs would vastly reduce unnecessary visits, safeguarding critical resources for optimizing quality MC service delivery.

As a national MC implementing partner, the ZAZIC consortium, led by the University of Washington's International Training and Education Center for Health (I-TECH) and local implementing partners, The University of Zimbabwe (UZ), Zimbabwe Association of Church-related Hospitals (ZACH), and Zimbabwe Community Health Intervention Research Project (ZiCHIRe), works in partnership with the Zimbabwe Ministry of Health and Child Care (MoHCC) to implement MC services in 13 districts [41]. Ensuring follow-up visit attendance, a routine MC program indicator, stretches scarce ZAZIC resources in an environment of few AEs: AE rates within both National and ZAZIC MC programs are less than 0.3% [33, 40]. Therefore, we implemented a randomized control trial (RCT) in two peri-urban clinics in

Zimbabwe to test two-way texting (2wT) between patients and providers, allowing men healing without complication to opt-out of their routine post-operative visits. RCT design details are available [42]. Clinical results from the RCT, published previously, identified AEs among 1.88% of 2wT arm participants and 0.84% of control arm men, suggesting that 2wT could safely reduce MC follow-up visits, reduce provider workload, and serve as a proxy for active surveillance [27].

In this mixed-methods study, we aimed to explore 2wT usability and acceptability among 2wT patients and healthcare providers focusing on understanding how providers and patients interacted with the 2wT system and obtaining insights into the obstacles and facilitators of system use. High usability and acceptability of 2wT is crucial for this technology to successfully expand across Zimbabwe and the region. Usability has been defined as the extent to which a product can be used by specified users to complete a task effectively, efficiently and with satisfaction in a specified setting [43–45]. More recent usability definitions also include concepts such as learnability, encouragement for use, minimizing errors, accessible to a wide range of individuals, and easy to maintain [46, 47]. Although *usability* may be applied differently across contexts, for the purposes of this paper, we focus on the perceived quality of the user experience interacting with the 2wT system [48], considering both client and provider input. For acceptability, we focused on the usefulness of the system and user perceptions of the importance of system functions. In contrast with usability, acceptance should not be reported by users, but should be demonstrated via evidence that users employ the technology for the intended purposes [49, 50]. Therefore, as evidence of *acceptability*, we explored client interactions with the system and their perceptions of its usefulness as an alternative to in-person follow-up care.

Evidence of high usability and acceptability is consequential for MC programs in Zimbabwe and elsewhere that are considering 2wT interventions to support follow-up. If the hybrid approach to integrating computer-automated messaging with human-to-human 2wT proved highly usable and acceptable, this digital health intervention could be applied to improve care and reduce provider workload in other follow-up care contexts. We hypothesized that 2wT acceptability and usability would be high, aiding program scalability and replicability.

## Theory of change

Ensuring 2wT usability and acceptability was critical to facilitating success from both provider and patient perspectives. Fig 1 maps how the 2wT intervention (program) aimed to influence individual factors (processes) in support of positive behavior change (individual outcomes) among male clients. 2wT targeted key individual-level constructs proven effective in previous HIV-related behavior change programs [51, 52]. First, 2wT providers enhanced post-operative counseling, encouraging self-efficacy to identify healthy healing [53] and control over the decision to return for in-person care if desired [54]. Then, via subsequent hybrid 2wT, providers connected with, and identified, men with potential AEs in need of provider triage and/or follow-up. These daily texts and swift individualized responses aimed to reinforce client motivation to respond [55]. Using theory to drive our intended effects, 2wT sought to empower men to correctly ascertain their healing progress and encouraged timely care seeking. We considered client engagement with the system as an indicator of usability and acceptability.

## 2wT technology overview

The 2wT system, the backbone of the intervention, was built using the Community Health Toolkit (CHT), an open source project which supports over 25,000 health workers across dozens of digital health activities. Apps built using the CHT work with or without internet

**Fig 1. 2wT theory of change.** 2wT theory of program (intervention) change.

connectivity, in any language, on basic phones, smartphones, tablets, and computers [56]. Since 2010 the CHT's core framework was shaped by a rigorous and well documented human-centered design process [57]} and is highly configurable. The overall CHT (https://communityhealthtoolkit.org/) supports the vast majority of the critical needs outlined by the WHO Classification of Digital Interventions [58]. Specifically, the 2wT system provides a model of four core WHO-promoted interventions at the patient and provider levels: 1) automated and patient-to-provider interactive messaging; 2) SMS-based triaging of clients by nurses (e.g. for referrals to care); 3) daily client monitoring via SMS; and 4) longitudinal patient records (potential AEs, AE follow up, referral confirmation) and reporting (e.g. client response rates). The 2wT features were designed to support a streamlined workflow and high quality MC services while generating data to monitor program delivery. The usability and acceptability of systems built using the CHT cannot be taken for granted; it is important that they be tested for specified purposes, with specified users, in particular use contexts.

## Methods

### 2wT intervention

Before full study implementation, message content was pre-tested for clarity of content and understandability in both Shona and English. Texts were reduced to 140 characters to ensure complete message content on multiple phone types and cell services. A pilot of 50 men assigned to texting intervention informed in-box modifications, SMS formatting, and message delivery optimizations (timing, frequency, language preferences). Pilot experience also identified wi-fi access via an internet dongle as the most resilient to fluctuations in electricity and cell networks. Automated messages were sent daily at 8am.

Implementation details of the prospective, un-blinded RCT study were described previously [27, 42]. In brief, ZAZIC follows all MoHCC protocols based on WHO guidelines including routine surgical MC follow-up on post-surgery days 2, 7 and 42 and adverse event (AE) management [59]. All MC care was provided free to clients from MoHCC. All men in the study (both 2wT and control arms) received all routine MC surgical care and both pre- and post-operative counseling at one of two public clinics: Seke South (peri-urban) or Norton (rural). However, in lieu of routine, in-person, post-operative visits, the 2wT clients received automated daily texts from days 1–13 and were asked to respond about their healing. Study texts

**Table 1. 2wT scripted message content.**

| Text purpose | English | Type |
|---|---|---|
| Day 0 Enrollment Confirmation | Thank you for participating in this study. | Auto |
| 8am: Days 1,3,4,5,6,8,9,10,11,12,13 | How are you? Are you experiencing any bleeding, swelling, pus, pain, redness or wound opening? Enter 1 = Yes, 0 = No and press send | Auto |
| 8am: Day 2 | It is Day 2. Remove the bandage. Any bleeding, swelling, pus, pain, redness or wound opening? Enter 1 = Yes, 0 = No and press send | Auto |
| If Client SMS "1" to the above | Which symptoms? Bleeding, swelling, pus, pain, redness, wound opening, or something else? | Manual |
| If they say they have a suspected AE | Please return to the clinic. If you would like a healthcare provider to call you, beep us. Otherwise, text us your question here | Manual |
| If they stop responding to SMS | Please seek VMMC follow-up care at the clinic | Manual |
| 8am: Day 7 | It is Day 7. Are you experiencing any bleeding, selling, pus, pain, redness or wound opening? Enter 1 = Yes, 0 = No and press send | Auto |
| 8am: Day 13, msg 1 | Thank you for participating in the study. Please return to the clinic tomorrow for your short Day 14 visit and to receive your airtime | Auto |
| 8:02am: Day 13, msg 2 | How are you? Are you experiencing any bleeding, swelling, pus, pain, redness or wound opening? Enter 1 = Yes, 0 = No and press send | Auto |
| Day 14 | Thank you for participating. Today is Day 14. Please return to clinic today for review and to receive airtime. | Auto |

are shown in Table 1. Daily responses were simple: only a single-digit response was required if the man believed he was healing well ("push '0' and hit send"), and no further action was taken. If a 2wT MC client responded with suspicion of an AE ("push '1' and hit send"), an MC nurse interacted with the client via SMS or phone, as requested. Clients chose the language of the automated texts at enrollment (English or Shona) but could engage the nurse in text communication using a mix of languages or common texting abbreviations. Clients were asked to return to the clinic if referred or desired. Follow-up communication at any time could be initiated by a client or the nurse. Participants in both arms were asked to return to the clinic on Day 14 for a study-specific review. No incentive was provided to study participants at enrollment. In recognition that sending texts in Zimbabwe costs approximately $0.05 [60], all study participants received a $5 phone card on Day 14 in appreciation for their participation.

## Quantitative methods

Of the 721 men in the RCT, 362 men were assigned to texting and included in this analysis. On day 14, a self-administered questionnaire was implemented with a subset of 100 of the 362 2wT MC clients (28%) to gauge satisfaction, acceptability, usability, and ascertain suggestions for intervention improvement. Responses of these 100 men were entered and frequencies explored in Excel for usability results. Data from all 362 are used to assess acceptability and usability as reflected in the system response rates and how the system was employed by the clients. Age and wage have plausible, *a-priori* relationships with usability and acceptability. Therefore, to determine whether the intervention was equally beneficial to all men, we explored the relationships between age and daily wage on both the number of text responses and potential AE texts among all 362 2wT texting arm men. The outcome variable, number of text responses, was a continuous measure of the number of daily texts to which a client responded, ranging from 0–13. The outcome variable, potential AE text, was a dichotomous variable defined as a client sending one or more potential adverse event responses to the daily text versus none. Both age and wage were categorized into quartiles. For daily wage, groups were: $0; 0-$5; $5-$15; and $16+. For age in years, categories were: 18–20; 21–24; 25–29; and 30 and above. We used linear regression for total text responses and logistic regression for potential AE text responses. Robust standard errors were used for all models. Coefficients and

odds ratios (OR) were presented for linear and logistic regression models, respectively. Multivariate models included both wage and age variables. STATA 15.1.0 (StataCorps, College Station, TX) was used for statistical analysis.

## Qualitative methods

We conducted brief (about 20 minutes) key informant interviews (KIIs) with eight clinicians involved in MC service delivery and 2wT-based follow-up to gauge acceptability; satisfaction; identify facilitators and barriers to program success; and ascertain suggestions for intervention improvement. Interviews were conducted at both 2wT intervention clinics: five interviews at Seke South and four at Norton. KIIs were audio recorded and transcribed. Thematic analysis [61] was employed, guided by a process of open coding based on anticipated responses suggested by the KII guide. Atlas.Ti software was used to create a spreadsheet of key themes, perceived barriers, and suggested facilitators to the program from KIIs. Usability questionnaires from the subset of 100 2wT texting men also contained several open-ended questions to ascertain perceptions of system use; those results were aggregated and grouped into themes using Excel.

## Ethics

This study was approved by the Medical Research Council of Zimbabwe (MRCZ) and the University of Washington, Seattle, USA, Internal Review Board. All subjects, clients and healthcare workers, received comprehensive information regarding their voluntary participation in the study and signed a written informed consent prior to study enrollment.

## Results

### Quantitative

Study enrollment occurred from June 18, 2018 to February 11, 2019; follow-up ended March 13, 2019. RCT enrollment, assignment, follow-up, and analysis of clinical outcomes were previously detailed [27]. The RCT enrolled 721 men and randomized them 1:1 to intervention (2wT) and control (routine): 362 (50%) were in the 2wT arm (Table 2) and included in this analysis of usability and acceptability. More than half (57%) of the 362 men responded to their daily text when it arrived at 8am, with 86% of responses within 4 hours. Over 92% responded to at least one daily text, showing high system acceptance. Response rates varied over time: approximately 70% of men responded on Day two while half responded on Day 13. Potential AE responses peaked on Day 2 and diminished thereafter (Fig 2). The nurse had 30 phone interactions with clients: 19 "call me back" requests were responded to and 11 calls were made to the nurse, directly, with a concern. Over the course of the intervention, clients sent over 1200 free text SMS to the 2wT nurse.

Closed-ended client survey responses suggest that most clients felt confident, comfortable, satisfied, and safe with SMS follow-up (Table 3). Few noted challenges with texting or were concerned about their wound healing without in-person review. Clients also felt that 2wT saved them time and money. As compared to alternative texting schedules, men preferred one daily text (76%), and they preferred that the text arrive in the morning (86%).

Both age and wage were associated with increased text response rates among 2wT men in univariate models (Table 4). On average, men from ages 21–24, 25–29, and 30+ sent an SMS on 2 more days than their younger peers, ages 18–20. Men with daily wages of $5-$15 or more than $15 sent an average of 1 or 2 more messages, respectively, over the follow-up period than those without income. In the multivariate model, age remained significantly associated with

**Table 2. Characteristics of 2wT arm men.**

| Characteristic | | 2wT | Full study |
|---|---|---|---|
| | | N = 362 (%) | N = 721 |
| Age category (years) | | | |
| | 0–20 | 92 (25.4) | 207 (28.7) |
| | 21–24 | 90 (24.9) | 178 (24.7) |
| | 25–29 | 75 (20.7) | 133 (18.5) |
| | 30+ | 105 (29.0) | 203 (28.2) |
| Site | | | |
| | Seke South | 300 (82.9) | 594 (82.4) |
| | Norton | 62 (17.1) | 127 (17.6) |
| Language selected for 2wT daily message | | | |
| | English | 154 (42.5) | 281 (39.0) |
| | Shona | 208 (57.5) | 440 (61.0) |
| Cell company | | | |
| | Econet | 321 (88.7) | 636 (88.2) |
| | Other | 41 (11.3) | 85 (11.8) |
| Daily wage category ($USD) | | | |
| | 0 | 108 (29.8) | 254 (35.5) |
| | >0–5 | 86 (23.8) | 155 (21.7) |
| | >5–15 | 98 (27.1) | 179 (25.0) |
| | >15 | 70 (19.3) | 128 (17.9) |

increased number of daily SMS responses. Although daily wage of $5-$15 compared to no income was associated with an increased likelihood of sending a potential AE message, this relationship was not significant when controlling for age in the multivariate model. Age was not associated with the likelihood of sending a potential AE text.

## Qualitative

**2wT clients.** For the open-ended responses among the 100 men who completed the usability survey on day 14, 45 responded (45%) to the question on formatting improvements; almost all responses suggested no change. Of the 42 men who responded about improved message content, the vast majority also suggested no change. However, addition of the day number (e.g., "this is the day 1 text") was suggested while a few others noted more information on the terms, e.g., "puss," should be explained. For the question on how to improve SMS follow-up, most responded that they liked the system as it is, but several noted that they would like to receive a call if they did not respond by text. Two men requested to be given airtime on Day 0. Overall, most men responded similarly to these participants who wrote, "*messages reduce time and if there is a problem you are given time to communicate. It's just an efficient way of communicating.*" Another client expressed that, "*the follow-up process is good because you can do what you want and also can make a monetary saving by not visiting the clinic.*"

**Healthcare workers.** Additional insight from the eight KIIs conducted after the completion of the full study suggested several themes to inform 2wT scale-up. Overall, the healthcare workers felt favorably towards 2wT, as men "*don't like coming for follow-up review visits.*" Another KI revealed that, "*in VMMC we have challenges with adult clients, they don't come [back for reviews], the numbers are not as you would want them to be.*" As a result, 2wT was welcomed by the HCWs, one of whom stated:

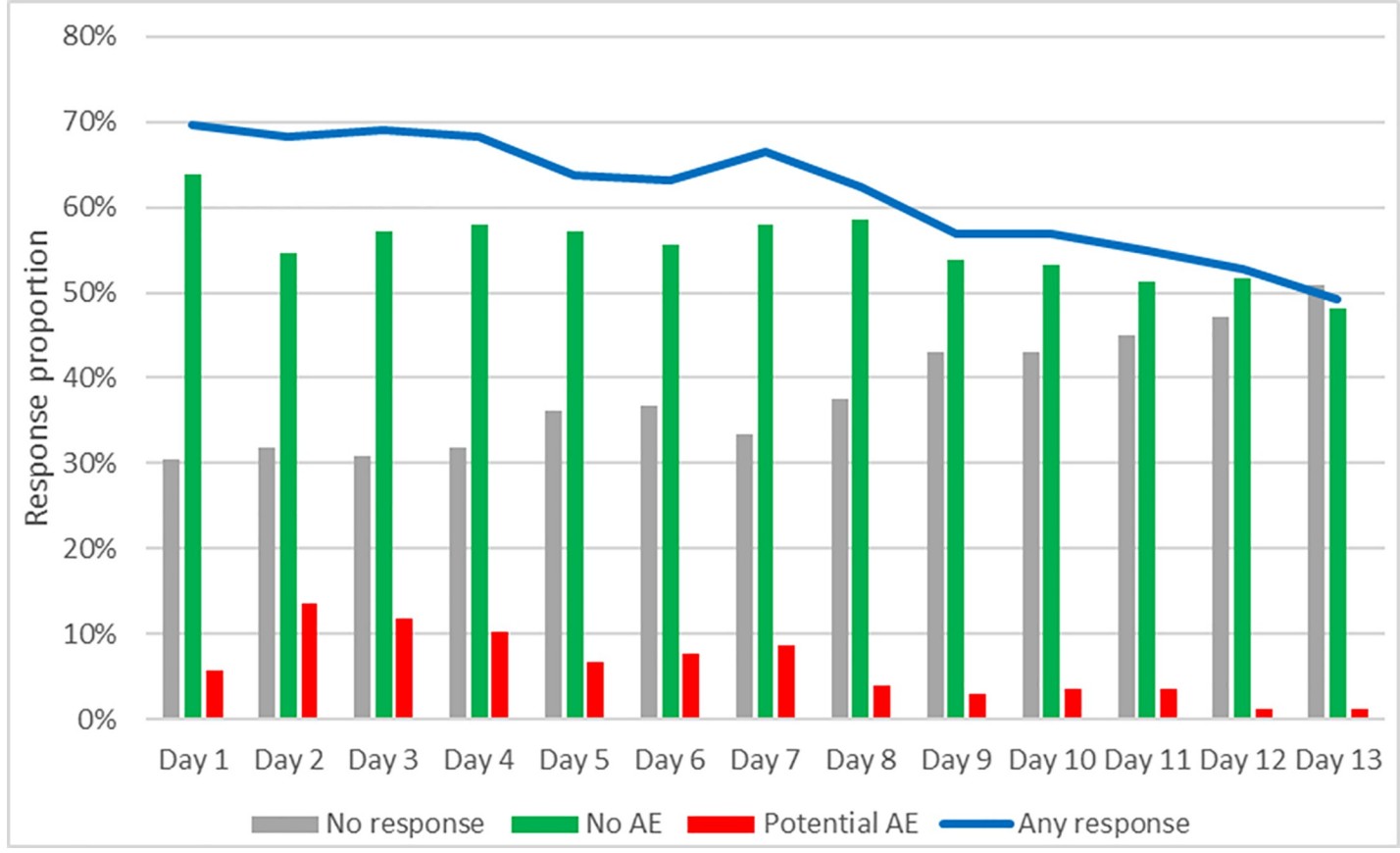

**Fig 2. Response rates over time, by response type.**

*We are looking forward to positive results that will lessen the burden on the service provider and improve post-operative quality care on the circumcised client and avoid any AE because of the on-going communication and feedback coming from the client.*

The KIs believed that 2wT reduced their overall workload. During the study, they "*only did a few reviews since clients were reviewed via cellphones.*" Scaling up 2wT with fewer post-operative reviews could lead to "*reducing burnout on health care workers. If you don't go for reviews it means you have enough time for other duties at the clinic.*" Several clinicians echoed the opinion that that 2wT could "*reduce the demand for staff to be going out and cut on costs for going to the field looking for clients when doing reviews.*"

KIs felt the 2wT intervention made the men partners in their healing process and "*gave the patients responsibility to cater for their health.*" It also gave the clinicians more confidence in their patients and "*actually helped us realise that men are actually more capable of taking care of themselves.*" Another KI noted that the intervention empowered men to be more vigilant in wound care.

"*We used to doubt that clients could manage themselves but because of the study we realized that people were responsible for their own health. . . They were empowered. They become part and parcel of the programme.*"

**Table 3. Responses on 2wT acceptability and usability from 2wT clients.**

| (N = 100) | Mean Likert score |
|---|---|
| I was clear on when to SMS, call, or return for care | 4.68 |
| I felt safe with the daily SMS | 4.63 |
| I was comfortable with this approach to MC follow-up care | 4.59 |
| I understood the SMS process well | 4.57 |
| I felt prepared to remove my bandages on Day 2 | 4.5 |
| I felt confident that I would get the care I needed at the clinic | 4.31 |
| I felt that I could come to the clinic when I wanted | 4.14 |
| I understood the signs of poor healing | 4.05 |
| I want fewer SMS | 3.05 |
| I wanted more SMS on signs of adverse events | 2.83 |
| I wanted help to remove my bandages on Day 2 | 2.73 |
| I was worried about my wound healing | 2.56 |
| I had challenges sending SMS | 2.33 |
| I had challenges receiving SMS | 1.89 |
| I wanted more in-person follow-up | 2.42 |
| I am worried about my privacy receiving these messages | 2.77 |
| No mandatory follow-up visits saved me money | 4.35 |
| No mandatory follow-up visits saved me time | 4.35 |
| I feel satisfied with the SMS follow-up | 4.61 |
| I would recommend SMS follow-up to my friends | 4.50 |

KIs also expressed several advantages specifically for clients. The KIs thought that 2wT would save men money "*by reducing transports costs. And also they were not phoning which means they were only using texts which cost only 5 cents.*" They also felt that clients saved time, as "*most of our clients are self-employed so the time wasted in returning to the clinic was reduced which was positive to the clients as well as to the service provider.*" When asked if they were

**Table 4. Factors associated with texting response rate and potential AE texts.**

| N = 362 | Total SMS responses† Coefficient (95% CI) | | Potential AE text†† OR (95% CI) | |
|---|---|---|---|---|
| | Univariate | Multivariate | Univariate | Multivariate |
| Age (years) | | | | |
| 0–20 | ref | ref | ref | Ref |
| 21–24 | 2.24*** (1.07–3.41) | 2.06*** (0.87–3.25) | 1.64 (0.90–2.98) | 1.63 (0.88–3.01) |
| 25–29 | 1.83** (0.59–3.08) | 1.49* (0.15–2.84) | 1.47 (0.79–2.76) | 1.39 (0.71–2.74) |
| 30+ | 2.38*** (1.25–3.51) | 1.91** (0.56–3.32) | 1.64 (0.92–2.92) | 1.66 (0.85–3.23) |
| Wage ($) | | | | |
| 0 | ref | ref | ref | ref |
| 0–5 | 0.97 (-2.00–2.15) | 0.81 (-0.36–1.98) | 1.54 (0.86–2.74) | 1.48 (0.82–2.68) |
| 5–15 | 1.39* (0.26–2.53) | 0.75 (-0.47–1.98) | 1.84* (1.05–3.22) | 1.56 (0.86–2.89) |
| 15+ | 1.93** (0.71–3.15) | 1.08 (-0.32–2.38) | 1.11 (0.60–2.07) | 0.89 (0.44–1.81) |

*p<0.05

**p<0.01

***p<0.001.

† Results from linear regression.

†† Results from logistic regression.

concerned that men would ignore signs of AEs, one KI summed bluntly: *"VMMC is a very sensitive area, men are worried about their organs. If they have got a problem they react, so there won't be any problems that they would ignore and not seek help."*

Lastly, clinicians shared several suggestions for 2wT scale-up. First, clients need clear instructions and post-operative counseling on wound care, bandage removal, and how to text to be able to safely use 2wT. Furthermore, clients needed reminders that 2wT was not for after-hours or emergency care. Several KIs noted potential issues for more rural areas with less reliable cell network or consistent electricity. One KI emphasized the need to maintain active tracing for non-responder, as provided currently for those who do not return to care. Lastly, although few clients noted financial challenges, four KIs suggested providing airtime on Day 1.

## Discussion

The results from mixed-method assessment of 2wT usability and acceptability strongly suggest that the hybrid automated/human-to-human SMS system is both acceptable and easy to use, complementing the clinical outcome data that showed 2wT between providers and clients helped ensure healthy wound healing and reduced unnecessary in-person visits [27]. The automated 2wT daily messages reminded clients of the signs of poor healing and encouraged wound observation for the most critical 13 days after MC, the period during which 95% of AEs occur [62]. Clients thought 2wT provided a similar level of reassurance as routine visits, but without the added time or money incurred from an in-person review. Clinicians believed 2wT empowered men to engage in their own healing, assuring them that their clients were capable of identifying potential problems and seeking care when they wished rather than on a mandated schedule. 2wT would not entirely replace clinical care or oversight, but could reduce the burden of unnecessary care on overstretched healthcare workers and reduce client costs while encouraging visits for those who desire or require care. 2wT's usability and acceptability appears to have helped ensure success. There are several lessons learned for future 2wT within the MC context or other shorter-term follow-up situations that may benefit from this type of direct provider-to-patient SMS.

Although digital health interventions document challenges in retention and adherence [63], 2wT improved upon previous texting interventions in several substantial ways. First, the technology, itself, was designed with usability at the forefront. Although mobile health (mHealth) interventions are growing in popularity [64–70], many short message service (SMS)-based health promotion efforts blast pre-defined messages to many people simultaneously [71–73], removing patients' ability to communicate back with healthcare workers [74]. By contrast, designing technology to support or augment end user skills and capabilities, as we did by enabling clients to ask questions and initiate care via SMS, is recognized as a central feature of human-centered approaches to system design [57]. Second, two-way texting with a mix of automated and human-to-human messaging optimizes for individualized content, efficiency for staff, and rapid follow up for the fraction of cases with complications, enabling interactive communication between clinicians and patients [75, 76]. Third, the short 13-day 2wT intervention maximizes client adherence, while common barriers such as phone theft, damage, and change of phone numbers [77] are minimized over the brief follow-up period. Fourth, the 2wT MC intervention implemented core characteristics of mHealth excellence, including client accessibility and acceptance; low technology costs, effective local adaptation, strong stakeholder collaboration, and government partnership for sustained impact [78]. Lastly, due to the high proportion of routine MC visits rendered moot when men confirm healing well via SMS, 2wT for MC realizes immediate gains in efficiency, unlike texting for patient education or behavior change.

The familiarity and flexibility of SMS (human-to-human messaging) likely played a role in usability and acceptability. All enrolled clients received their initial text while still in the clinic with MC healthcare providers, providing a personal connection and immediate reassurance that the texting system worked. Post-operative counseling, enhanced with emphasis on expected text responses and the healing process, likely increased client confidence in their ability to recognize and report potential AEs via SMS and act on both information and guidance provided via text. The automated daily text messages also appeared to comfort clients, helping them feel attended to without the need for an in-person visit. Few men requested calls or called the nurse directly, likely suggesting that the men felt that their needs were well met via SMS. For providers, the automated daily SMS significantly reduced the burden on the study nurse as the system, itself, handled the majority of outgoing messages. This enabled the nurse to prioritize time and communication with the clients who suspected an AE or who, for any other reason, initiated a text conversation or requested a call. By reducing the burden on the nurse, and removing the need for further communication with clients who responded with "No AE," she was able to respond to many more clients than she would have been able to manage in person at the clinic.

This intervention increased male engagement in care, a recognized need in HIV programs. Despite the fact that men's uptake of healthcare services is lower than that of women [79], interventions that increase male involvement in HIV-related care are absent in much of SSA. Men are less likely to enter the HIV care cascade [80, 81], especially in rural areas [82], threatening HIV prevention efforts. mHealth interventions, like 2wT, show progress in increasing male engagement with care[83, 84]. 2wT appeared acceptable, feasible, and empowering for male clients [85], successfully garnering active participation from men in post-MC healing. 2wT in Zimbabwe appears beneficial to most adult men involved in the RCT, with minimal changes in effectiveness due to income or age. Older men were more likely to respond to the texts, but this small effect does not seem programmatically meaningful. Although clients were not given airtime until day 14, response rates of almost 93% suggest that men were able and willing to pay SMS costs up front. Clients likely paid an average of $0.05 per SMS, and were asked to respond to a daily text for a minimum combined cost of $0.65, or far less if they bought text bundles. Many men exchanged further texts with the nurse, suggesting that men were able and willing to pay these SMS costs in lieu of an in-person visit that would likely require travel time, transportation, and absence from work.

The results of the quantitative and qualitative analysis suggest several key improvements for 2wT at scale. First, the nurse managing the clients urged future scale-up interventions to clarify that 2wT is not for emergency or after-hours care. This could easily be completed by an automated response recommending alternatives for seeking care after hours. Similarly, at scale more nurses would be needed to rotate responsibility for client management via the 2wT system in anticipation of client communication requests and timing; further user interface optimizations could support these programmatic changes. Moreover, most mobile network operators offer 'reverse billed' SMS numbers that are free to clients and paid for by the sponsoring organization; while costly for a small study, this would be viable at scale. These steps may be critical for 2wT at scale to reduce the workload on the nurse managers and ensure continuous client care.

It is also worth noting several limitations to the 2wT intervention. The majority of study clients liven in an urban setting, and it remains to be investigated whether 2wT interventions may be similarly usable among rural populations. In settings where phone ownership remains low, the efficiency gains of 2wT may be limited to a subset of the total client base. For safety and verification, our study protocol also excluded any client who did not have a phone with him at the time of enrollment and who did not receive a confirmation message from the 2wT

system before leaving the clinic. 2wT at scale could opt for less strict inclusion criteria to enroll more clients and increase 2wT access; however, that could introduce patient care and verification concerns if hardware malfunctions, SIM card issues, poor electricity access, or poor cell networks jeopardized potential client contact. Consideration of these constraints or other challenges such as phone sharing were beyond the scope of the present 2wT study. Lastly, although we did not encounter high rates of refusal, there are privacy and stigma concerns among men opting for MC that affect follow-up rates. However, if men were allowed to access 2wT at any time during the healing period, without requiring 2wT enrollment at registration, 2wT could provide men with concerns an alternative pathway to access care, improving patient outcomes overall. Despite these limitations, and combined with the clinical results demonstrating safety [27], it is clear that 2wT warrants further exploration for replication and scale-up in other MC care contexts and settings.

## Conclusions

Our usability and acceptability results show that post circumcision 2wT interaction for follow-up care is highly useable and acceptable for clients and healthcare professionals, contributing substantially to the successful achievement of MC program efficiencies. By designing a hybrid workflow, we were able to integrate the efficiency benefits of automated messaging with the familiarity and flexibility of human-to-human conversational texting. While our usability and acceptability findings are grounded in the clinical and operational particulars of MC, our results are largely attributable to the software and hybrid 2wT workflow rather than to a specific disease or condition, contributing to the growing applicability of 2wT in healthcare interventions in developing countries, including SSA [86–89]. Similar automated/human-to-human 2wT merits further study in other care contexts, such as childhood diseases, respiratory infections, or post-operative care that could similarly benefit from intensive, direct provider-to-client communication.

Based on our mixed-methods findings, we would emphasize four characteristics of health programs that may benefit from the hybrid automated/human-to-human workflow: 1) facility-based follow-up care is perceived as burdensome to patients and/or providers; 2) most clients heal, adhere, or otherwise succeed with routine care and standard counseling, minimizing risk; 3) clients with complications or questions are likely to be able to self-care or triage safely and effectively with the support of a remote clinician; and 4) the follow-up period is brief enough to sustain engagement and minimize obstacles due to lost phones, changed phone numbers etc. Programs with these characteristics may be more amenable to a new kind of task-shifting, in which digital tools enable clients and their care givers to be more informed, engaged, and capable of self-care in situations that otherwise might have required a visit to a health facility.

The open-source nature of our intervention should aid efforts to replicate this intervention or adapt it for other settings and care contexts. However, 2wT adaptations, replications, and advances are not without risks and not without costs. Further research on hybrid 2wT interventions across diseases and care contexts, with attention to the implementation challenges that may limit scale-up, could further establish the benefits and limitations of this new approach to digitally supported self-care in the community.

## Supporting information

**S1 Data.**
(DOCX)

**S2 Data.**
(XLSX)

## Acknowledgments

The authors wish to acknowledge the countless contributions of the 2wT study team, Christina Mauhy, Mujinga Tshimanga, Wendy Mutepfe, and Patricia Tapiwa Gundidza, towards the success of this research study. We would also like to thank the Zimbabwe MC teams at Seke South and Norton clinics as well as Zimbabwe Ministry of Health and Child Care for their participation and collaboration on the 2wT effort. The authors would also like to thank the University of Washington / Fred Hutchinson Center for AIDS Research, an NIH funded program, for biostatistical support.

## Author Contributions

**Conceptualization:** Caryl Feldacker, Isaac Holeman, Scott Barnhart.

**Data curation:** Caryl Feldacker.

**Formal analysis:** Caryl Feldacker.

**Funding acquisition:** Caryl Feldacker.

**Investigation:** Caryl Feldacker, Vernon Murenje, Scott Barnhart, Mufuta Tshimanga.

**Methodology:** Caryl Feldacker, Isaac Holeman.

**Project administration:** Vernon Murenje, Sinokuthemba Xaba, Batsirai Makunike-Chikwinya, Mufuta Tshimanga.

**Software:** Isaac Holeman, Michael Korir, Bill Wambua.

**Supervision:** Caryl Feldacker, Vernon Murenje, Sinokuthemba Xaba, Batsirai Makunike-Chikwinya, Marrianne Holec, Mufuta Tshimanga.

**Validation:** Caryl Feldacker, Michael Korir, Scott Barnhart.

**Visualization:** Bill Wambua.

**Writing – original draft:** Caryl Feldacker, Isaac Holeman, Scott Barnhart.

**Writing – review & editing:** Caryl Feldacker, Vernon Murenje, Sinokuthemba Xaba, Michael Korir, Bill Wambua, Batsirai Makunike-Chikwinya, Marrianne Holec, Mufuta Tshimanga.

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
