## [Decision Letter · Decision Letter 0]

17 Feb 2020

PONE-D-19-27561

Usability and acceptability of a two-way texting intervention for post-operative follow-up for voluntary medical male circumcision in Zimbabwe

PLOS ONE

Dear  Dr Feldacker,

Thank you for submitting your manuscript to PLOS ONE. After careful consideration, we feel that it has merit but does not fully meet PLOS ONE’s publication criteria as it currently stands. Therefore, we invite you to submit a revised version of the manuscript that addresses the points raised during the review process.

We would appreciate receiving your revised manuscript by 30th March 2020. To enhance the reproducibility of your results, we recommend that if applicable you deposit your laboratory protocols in protocols.io, where a protocol can be assigned its own identifier (DOI) such that it can be cited independently in the future. For instructions see: http://journals.plos.org/plosone/s/submission-guidelines#loc-laboratory-protocols

We look forward to receiving your revised manuscript.

Kind regards,

Kwasi Torpey, MD PhD MPH

Academic Editor

PLOS ONE

2. Thank you for letting us know that Figures 1 and 3 in your submission contain copyrighted images. All PLOS content is published under the Creative Commons Attribution License (CC BY 4.0), which means that the manuscript, images, and Supporting Information files will be freely available online, and any third party is permitted to access, download, copy, distribute, and use these materials in any way, even commercially, with proper attribution. For more information, see our copyright guidelines: http://journals.plos.org/plosone/s/licenses-and-copyright.

a)    You may seek permission from the original copyright holder of Figures 1 and 3 to publish the content specifically under the CC BY 4.0 license.

Reviewers' comments:

Reviewer's Responses to Questions

**Comments to the Author**

1. Is the manuscript technically sound, and do the data support the conclusions?

Reviewer #1: Yes

Reviewer #2: Yes

Reviewer #3: No

2. Has the statistical analysis been performed appropriately and rigorously? 

Reviewer #1: Yes

Reviewer #2: Yes

Reviewer #3: No

3. Have the authors made all data underlying the findings in their manuscript fully available?

Reviewer #1: No

Reviewer #2: Yes

Reviewer #3: No

4. Is the manuscript presented in an intelligible fashion and written in standard English?

Reviewer #1: Yes

Reviewer #2: Yes

Reviewer #3: No

5. Review Comments to the Author

Reviewer #1: Important topic. However the way the mansucript reads currently, it requuires major edits to give it focus on the one objective of this study. To assess usability and acceptability.

Overall: Important topic and research results. The manuscript needs major editing to improve clarity.

Abstract

1. Objective that are not part of this paper are included. Need to focus only on what is included in the current manuscript. Other objectives of the overall study could be mentioned in the methods section.

2. Remove numbers on the aims – and they should only be two – according to the tittle: usability and acceptability of the 2wT

3. Conclusion: Findings on “men with concerns…” is not included in the results section of the abstract

Main Manuscript

Introduction

1. Lines 94 – 109 – should be part of the methods section

2. Lines 87 – 97 – should be in the setting section of the manuscript under methods

3. Lines 123- 124 – revise to indicate objective/aim of this study.

Methods

1. Theory of change section should be moved to the introduction section

2. Line 150 - 2wT Technology overview – reads like a descript of the intervention

3. Intervention – reads more like study procedures.

4. Line 176 – Only Shona and English were used in the testing, what about other languages like Ndebele. What was done if the participant did not understand any of those two languages (Shona and English)

5. Line 213 – Quantitative methods: Open-ended questions are part of qualitative research. Likert scale questions and demographics possible quantitative. It might be better in this section to put data collection from MC participants, and data collection from healthcare worker

6. Lines 231 – 236: Data analysis and the statistical analysis could be included in this section.

Results

1. Healthcare workers perspective – too long, and can be summarized highlighting the major findings

Discussion

1. Lines 410 – 415 – edited and be moved to study procedures or intervention description

2. Line 467 “it is clear that 2wT is warrants further” delete the word is

Reviewer #2: This manuscript addresses concerns regarding the utility of adherence to post-imperative follow-up visits after VMMC. The paper would be strengthened with better organization and more focus on the quantitative outcomes. Specific comments are below:

1) Line 67 – More recent data than 2017 are available for MCs done in SSA.

2) Line 77 – I would specify that when you are only referring to post-operative AEs, not intra-operative.

3) Line 81 – I would specify the post-operative recommendations recommended by Zimbabwe’s MOHCC.

4) Lines 94-108 – While this is important information, most of it should appear in the methods and results sections, rather than the introduction.

5) Line 124 – The authors refer to “Aim 3” without any context, so this does not make sense to the reader. I would recommend dropping the references to Aims here and throughout the paper.

6) Line 138 – There might be an error with the figures and tables, as Figure 1 is the CONSORT table, not a mapping of the intervention.

7) Lines 150-171 – This information should be incorporated into the background/introduction rather than the methods.

8) Lines 174-190 – This detail about pretesting is interesting but not critical to the reader to understand the piloting and refining that took place as part of this study. I would recommend deleting this section and reducing this text to a sentence or two.

9) Line 197-198 – This description of the control arm is confusing, as it appears that the intervention arm had the same follow-up schedule (additional visit on Day 14) I would recommend a clearer description of the arms, or an additional table if possible.

10) Line 223 – Specify that the wage groups are for daily wages earned.

11) Line 230 – It’s unclear if these are facilities, sub-districts, etc.

12) Line 255 – There is a typo—it says “participate” when it should be “participant”.

13) Line 270 – Inconsistent use of the oxford comma here and throughout.

14) Line 275 – This table would be easier to interpret if it was in descending order by Likert score or grouped into relevant categories.

15) Line 281 – I don’t see how this point is relevant to the broader findings of this study.

16) Line 293 – 369 – The qualitative findings are not compelling enough to add to the overall value of this paper. I would recommend either deleting them and focusing on the quantitative findings, or developing this section further to have more robust findings and conclusions.

17) Results section - I assume that this was previously described in more detail elsewhere, but a major gap in this manuscript is that the AE results by arm are not reported in the results and the findings in this paper do not engage with the overall findings from the main study outcomes. The reader would also need to know how many AEs were missed through the self-diagnosis.

18) Line 377 – Why is Day 13 most critical?

19) Line 384 – The content of these daily messages need to be presented somewhere in this manuscript.

20) Line 388 – I think you mean “substantial”, not “significant”.

21) Line 467 – It is impossible to know if 2wT warrants further explanation without knowing how effective it was to diagnose actual post-operative AEs by having the men self-diagnose.

22) Tables and Graphs – Several of the graphs lack titles, and the bar graph showing time of day would work better as a couple of sentences, rather than a figure.

Reviewer #3: 1. The manuscript is not organized. Methods are in different places and need to be concisely captured to clearly bring out the study design. Introduction lines 99 to 108 are the only section that tend to describe the design. This section should be moved to the methods section and the results therein should be in the results not methods.

2. Related to above, is the RCT the best design for this study? There is limited analysis that shows comparison of the two study arms. The randomization of 771 mentioned is not followed through in the analysis. Analyses follows N=100 and these seem to be in the intervention arm. Where is the rest of the sample?

3. How was the sample of 771 determined?

4. The study outcomes are not clearly stated. What is the primary outcome and what are the secondary outcomes? These must be measurable in both arms. You have text responses and AEs for instance. Are text responses measured in the control arm?

5. Because study outcomes are not clear as they relate to both arms analyses herein may not be optimum for the design. It suffices to say that analyses must show study outcomes for both outcomes. You could explore difference in difference analyses for example but even that needs to be informed by the data and theory of change.

6. The theory of change is also not clear and requires recasting. Graphical presentation of this showing variables being measured in the study and direction of change with clear outcomes as mentioned above is important for clarity. As it stands it mentions even variables that are not presented in the analyses.

7. The whole section on technology is not necessary. This should be rolled into the intervention description and need not be this long.

8. You have under methods a sections on quantitative and qualitative. These two should be rolled out into design description and should bring out design aspects more than tools description.

9. Related to above, the qualitative narrative suggests that coding was informed by the study guide. Codes should come from themes in the data not the guide. In this section you also mention that analysis is of a subset of the sample, why? This is not explained.

10. The statistical analysis is not clear on the comparison needed for an RCT. Why are robust errors used in the analysis?

11. Ethics section should mention approval references from the ethics boards mentioned.

12. In the results table 1 compares intervention arm to the total study. Why? You need to compare the two arms to determine control variables.

13. Subsequent tables 2 and 3 only analyze data for the intervention arm. What was the need for the control. AEs outcome should be compared between the two arms. Where is analyses presented?

14. Interpretation of ORs appears problematic. When Cis cross 1 results are not statistically significant. This does not appear to be applied in these analyses.

6. PLOS authors have the option to publish the peer review history of their article (what does this mean?). If published, this will include your full peer review and any attached files.

Reviewer #1: Yes: Limakatso Lebina

Reviewer #2: No

Reviewer #3: No

---

## [Author Response · Author response to Decision Letter 0]

12 Apr 2020

To the editor and reviewers,

Thank you very much for the thorough review. The authors have revised the manuscript in accordance with the suggested changes and believe that the manuscript is stronger as a result. We prepared a point by point response to the reviewers below. Our responses are in italics with line and page number to indicate new text or changes. 

We appreciate your consideration and support for this important contribution to the literature.

Best,

Caryl Feldacker on behalf of all authors 

Reviewer 1: 

Abstract

1. Objective that are not part of this paper are included. Need to focus only on what is included in the current manuscript. Other objectives of the overall study could be mentioned in the methods section. Revised

2. Remove numbers on the aims – and they should only be two – according to the tittle: usability and acceptability of the 2wT Revised

3. Conclusion: Findings on “men with concerns…” is not included in the results section of the abstract This is addressed in the results. Men interacted with the nurse via individual, tailored messages. There were over 1500 messages exchanges, signaling that the men were interacting with the nurse over their healing concerns. The phrasing of the abstract was updated for added clarity. 

Main Manuscript

Introduction

1. Lines 94 – 109 – should be part of the methods section Much of this was deleted or moved as the clinical results were published. 

2. Lines 87 – 97 – should be in the setting section of the manuscript under methods. This information in the introduction sets the background for the RCT. We shortened it but prefer to leave it in the introduction. We did note the study clinics within the 2wT intervention information on lines 170-171. 

3. Lines 123- 124 – revise to indicate objective/aim of this study. Updated in lines 97-101. 

Methods

1. Theory of change section should be moved to the introduction section. Moved and clarified.

2. Line 150 - 2wT Technology overview – reads like a descript of the intervention. This section was revised and streamlined to focus more on the technology background and its core components for this intervention. The 2wT system was the backbone of the intervention. 

3. Intervention – reads more like study procedures. The study was the 2wT intervention, itself. We tried to rewrite this section in accordance with reviewer suggestions, and we noted only the core components of the intervention for the texting men. 

4. Line 176 – Only Shona and English were used in the testing, what about other languages like Ndebele. What was done if the participant did not understand any of those two languages (Shona and English). In this region of Zimbabwe, Shona and English predominate. We did not encounter anyone who noted a desire for automated texts in another language. Clients could send and respond to texts using any language, something that we now noted on lines 176-178. 

5. Line 213 – Quantitative methods: Open-ended questions are part of qualitative research. Likert scale questions and demographics possible quantitative. It might be better in this section to put data collection from MC participants, and data collection from healthcare worker. All information on the open ended questions from the client usability survey was moved to qualitative. 

6. Lines 231 – 236: Data analysis and the statistical analysis could be included in this section. Changed and all moved to the quantitative methods section. 

Results

1. Healthcare workers perspective – too long, and can be summarized highlighting the major findings. The quotes were shortened as was the section. For scale-up, it is critical to include enough information from the HCWs to inform considerations moving from research to practice. 

Discussion

1. Lines 410 – 415 – edited and be moved to study procedures or intervention description. Moved

2. Line 467 “it is clear that 2wT is warrants further” delete the word is. Deleted

Reviewer #2: This manuscript addresses concerns regarding the utility of adherence to post-imperative follow-up visits after VMMC. The paper would be strengthened with better organization and more focus on the quantitative outcomes. Specific comments are below:

1) Line 67 – More recent data than 2017 are available for MCs done in SSA. Updated. Line 62 

2) Line 77 – I would specify that when you are only referring to post-operative AEs, not intra-operative. As much of the literature refers to both or do not specify, I am referring to all AEs here. 

3) Line 81 – I would specify the post-operative recommendations recommended by Zimbabwe’s MOHCC. Specified line 77

4) Lines 94-108 – While this is important information, most of it should appear in the methods and results sections, rather than the introduction. Largely moved to methods

5) Line 124 – The authors refer to “Aim 3” without any context, so this does not make sense to the reader. I would recommend dropping the references to Aims here and throughout the paper. Removed throughout. 

6) Line 138 – There might be an error with the figures and tables, as Figure 1 is the CONSORT table, not a mapping of the intervention. All figure and table numbers revised. 

7) Lines 150-171 – This information should be incorporated into the background/introduction rather than the methods. This section now comes before the methods. It is difficult to determine the best placement for this critical section as evidenced by the conflicting advice from the reviewers. We are open to placing these sections in the most logical order according to the reviewers and the Editor. 

8) Lines 174-190 – This detail about pretesting is interesting but not critical to the reader to understand the piloting and refining that took place as part of this study. I would recommend deleting this section and reducing this text to a sentence or two. Reduced and placed in the 2wT intervention section of the methods. Line 156 -162. 

9) Line 197-198 – This description of the control arm is confusing, as it appears that the intervention arm had the same follow-up schedule (additional visit on Day 14) I would recommend a clearer description of the arms, or an additional table if possible. This section was revised for clarity. Please see lines 163-181. 

10) Line 223 – Specify that the wage groups are for daily wages earned. This is now clarified throughout. 

11) Line 230 – It’s unclear if these are facilities, sub-districts, etc. Clarified in line 168, “Interviews were conducted at both 2wT intervention clinics: five interviews were conducted at Seke South and four at Norton”

12) Line 255 – There is a typo—it says “participate” when it should be “participant”. Corrected, thank you. I wish my editing were as good as yours!

13) Line 270 – Inconsistent use of the oxford comma here and throughout. We have done our best to correct this throughout. 

14) Line 275 – This table would be easier to interpret if it was in descending order by Likert score or grouped into relevant categories. Reordered by Likert score. Thank you!

15) Line 281 – I don’t see how this point is relevant to the broader findings of this study. Age was positively related to increased number of SMS responses – or older men were more likely to respond more frequently. We think this is important to understands who might find the system more acceptable especially as the age pivots to focus MC on those 15-29. 

16) Line 293 – 369 – The qualitative findings are not compelling enough to add to the overall value of this paper. I would recommend either deleting them and focusing on the quantitative findings, or developing this section further to have more robust findings and conclusions. We understand that different readers will have different preferences and find value in specific components of any research. We believe your point is valid; however, it was this depth of information that helped the MoHCC review the 2wT proposal for expansion favorably. We believe that hearing what the men said, in their own word, is important to understand how the men used the system. Likewise, we believe that hearing from the HCWs, themselves, about why they think 2wT succeeded and how to move from research to practice is invaluable. We did shorten this section, including making the quotes more succinct. We hope this will make them more compelling. 

17) Results section - I assume that this was previously described in more detail elsewhere, but a major gap in this manuscript is that the AE results by arm are not reported in the results and the findings in this paper do not engage with the overall findings from the main study outcomes. The reader would also need to know how many AEs were missed through the self-diagnosis. The previous paper on the clinical outcomes focused only on AE ascertainment, overall client safety, and workload. It is available here: https://journals.lww.com/jaids/Fulltext/2020/01010/Reducing_Provider_Workload_While_Preserving.3.aspx/ While we have to give some information about that aspect of the clinical trial, we are trying very hard to focus only on usability and acceptability of the system, itself, in this paper. We did add one sentence to the background to make this more clear. We do not know how many AEs were missed through self diagnosis or through simple healing before identification; however, as there was only 1 previously undetected AE identified at the Day 14 visit (within the control arm), we noted within that clinical paper that we are confident that there was not a huge backlog of unidentified AEs that went undetected over the course of the study. In the introduction, we now comment, “Clinical results from the RCT, published previously, identified AEs among 1.88% of 2wT arm participants and 0.84% of control arm men, suggesting that 2wT could safely reduce MC follow-up visits, reduce provider workload, and serve as a proxy for active surveillance [27]. However, the other reviewer suggested taking out all previous findings. We hope the readers will review the clinical outcomes for more information on these results. 

18) Line 377 – Why is Day 13 most critical? Our previous study that informed this aspect of the RCT found that 95% of AEs happen before day 13: https://www.ncbi.nlm.nih.gov/pubmed/30192816. That line is now augmented with the citation, “The automated 2wT daily messages reminded clients of the signs of poor healing and encouraged wound observation for the most critical 13 days after MC, the period when 95% of AEs occur” on line 331. 

19) Line 384 – The content of these daily messages need to be presented somewhere in this manuscript. Excellent suggestion. Added a table 1. 

20) Line 388 – I think you mean “substantial”, not “significant”. Changed. 

21) Line 467 – It is impossible to know if 2wT warrants further explanation without knowing how effective it was to diagnose actual post-operative AEs by having the men self-diagnose. Changed this to reference the clinical results demonstrating safety, “Despite these limitations, and combined with the clinical results demonstrating safety [27], it is clear that 2wT warrants further exploration for replication and scale-up in other MC care contexts and settings. lines 418-420. 

22) Tables and Graphs – Several of the graphs lack titles, and the bar graph showing time of day would work better as a couple of sentences, rather than a figure. Graph replaced and language changed: “More than half (57%) of the men responded to their daily text when it arrived at 8am, with 86% of responses within 4 hours.” Lines 230-232

Reviewer #3: 

1. The manuscript is not organized. Methods are in different places and need to be concisely captured to clearly bring out the study design. Introduction lines 99 to 108 are the only section that tend to describe the design. This section should be moved to the methods section and the results therein should be in the results not methods. Thank you for this comment. We reorganized this manuscript and hope it will read more logically for readers now. We are trying to briefly present the methods from the overall RCT (clinical results and design details already published here: https://journals.lww.com/jaids/Fulltext/2020/01010/Reducing_Provider_Workload_While_Preserving.3.aspx#O8-3-2), not overlap, and give enough context for these critical study results on usability and acceptability. We need to describe part of the overall study to orient the reader, but the RCT methods, overall, are background to this exploration of the system response. We did reorganize and hope that it is more clear now. 

2. Related to above, is the RCT the best design for this study? There is limited analysis that shows comparison of the two study arms. The randomization of 771 mentioned is not followed through in the analysis. Analyses follows N=100 and these seem to be in the intervention arm. Where is the rest of the sample? The RCT was already conducted and clinical outcomes published. The design of the RCT was published here and cited in the background of the paper: (https://trialsjournal.biomedcentral.com/articles/10.1186/s13063-019-3470-9). Of the 712 randomized 1:1 in the RCT, 362 were in the texting arm (Table 2). We clarified in the quantitative methods on lines 187-196: “Of the 721 men in the RCT, 362 men were assigned to texting and included in this analysis. On day 14, a self-administered questionnaire was implemented with a subset of 100 of the 362 2wT MC clients (28%) to gauge satisfaction, acceptability, usability, and ascertain suggestions for intervention improvement. Responses of these 100 men were entered and frequencies explored in Excel for usability results. Data from all 362 are used to assess acceptability and usability as reflected in the system response rates and how the system was employed by the clients. Age and wage have plausible, a-priori relationships with usability and acceptability.” In Tables 2 and4, we now note n=362 while Table 3 notes n=100. We hope this makes it more clear for the reader. 

3. How was the sample of 771 determined? The methods of the RCT are presented in the JAIDS publication and are not the focus of this paper. We do not want to replicate previously published content. In brief, For the clinical component (safety as measured by AE rates), we determined the sample size based on hypothesis testing for non-inferiority to examine the outcome of interest: AE rate (moderate or severe) occurring ≤ 14 days. The non-inferiority margin, based on statistical and clinical considerations, is the maximum difference between the rate of AEs ≤ 14 days in the control and 2wT arms where we would conclude that 2wT is not inferior to routine care. We set that margin to 1.6%, which would create a non-inferiority cutoff of ≤ 2% AEs in 2wT. The cutoff for the proportion of AEs to define 2wT non-inferiority was set at a conservative 2% of AEs to match the lower bound of AE rates reported in previous rigorous studies and because an AE rate of 2% is regarded as a commonly used standard of VMMC safety, including in Zimbabwe. Including a possible 10% LTFU, 361 participants per arm provides 90% power at alpha of 2.5% to detect AEs above the non-inferiority margin of 1.6%, finding 2wT non inferior if we rule out an AE rate > 2%. We found that AE rates were not different per arm but that we could not rule out non inferiority as we ascertained more AEs using 2wT. We believe, and asserted in that paper, the 2wT acted as a proxy for active surveillance. 

4. The study outcomes are not clearly stated. What is the primary outcome and what are the secondary outcomes? These must be measurable in both arms. You have text responses and AEs for instance. Are text responses measured in the control arm? The control arm received no texts, so we cannot compare usability or acceptability across arms. The main clinical outcomes, that compare across arms, was published previously. We clarified our aim in the introduction, paragraph 4: “In this mixed-methods study, we aimed to explore 2wT usability and acceptability among 2wT patients and healthcare providers focusing on understanding how providers and patients interacted with the 2wT system and offering insights into the obstacles and facilitators of system use.” Earlier in the introduction, as background, we note the previous RCT results, “Therefore, we implemented a randomized control trial (RCT) in two peri-urban clinics in Zimbabwe to test two-way texting (2wT) between patients and providers, allowing men healing without complication to opt-out of their routine post-operative visits. The trial design details are available {Feldacker, 2019 #928}. Clinical results from the RCT, published previously, identified AEs among 1.88% of 2wT arm participants and 0.84% of control arm men, suggesting that 2wT could safely reduce MC follow-up visits, reduce provider workload, and serve as a proxy for active surveillance [27]”

5. Because study outcomes are not clear as they relate to both arms analyses herein may not be optimum for the design. It suffices to say that analyses must show study outcomes for both outcomes. You could explore difference in difference analyses for example but even that needs to be informed by the data and theory of change. The theory of change applies only to the intervention group. The control arm received routine VMMC care and was not exposed to the intervention nor allowed or offered texting as a form of follow-up. 

6. The theory of change is also not clear and requires recasting. Graphical presentation of this showing variables being measured in the study and direction of change with clear outcomes as mentioned above is important for clarity. As it stands it mentions even variables that are not presented in the analyses. The theory of change has been rewritten for clarity and better written description of the figure, lines 120-132. We note how each one of the outcomes was measured in the overall approach. This figure shows the intervention theory of change which encompasses the clinical outcomes. Interaction with the system in the manner depicted signifies usability and acceptability of the system, overall. 

7. The whole section on technology is not necessary. This should be rolled into the intervention description and need not be this long. This section was shortened and the figure removed. However, we feel like this brief introduction and overview of the technology is critical to the reader’s understanding of the usability and acceptability of the technology. Informing and potentially adapting the technology, itself, is the main focus of the usability and acceptability study. 

8. You have under methods a sections on quantitative and qualitative. These two should be rolled out into design description and should bring out design aspects more than tools description. Respectfully, we disagree and believe that these methods belong in the current section of the paper. We are open to moving them if also suggested by the Editor. 

9. Related to above, the qualitative narrative suggests that coding was informed by the study guide. Codes should come from themes in the data not the guide. In this section you also mention that analysis is of a subset of the sample, why? This is not explained. Coding was informed by the study guide that contained questions of interest about the 2wT intervention. As a process of open probing in response to the informants also occurred, we followed themes that stemmed from the guide-prompted questions as well as from the free response of the informants. As noted in the quantitative methods, a subset of 100/362 men were asked to complete a usability questionnaire at the Day 14 visit. The open-ended responses of those 100 respondents form the qualitative data from clients. 

10. The statistical analysis is not clear on the comparison needed for an RCT. Why are robust errors used in the analysis? Some statisticians, including the one who worked on this paper, believe that logistic regression is heteroskedastic by nature, requiring robust standard errors. Others would use robust standard errors as we have data on only a few predictors. Some would disagree, but the robust standard errors make the CIs larger, and therefore more conservative. We chose to be more conservative with our approach. 

11. Ethics section should mention approval references from the ethics boards mentioned. I am unclear. In the text, we note, “study was approved by the Medical Research Council of Zimbabwe (MRCZ) and the University of Washington, Seattle, USA, Internal Review Board.” These are the approvals we received. 

12. In the results table 1 compares intervention arm to the total study. Why? You need to compare the two arms to determine control variables. I did not want to duplicate previously published material. The first publication on the clinical outcomes contains these analyses. This paper is only about the 2wT (intervention) men and, therefore, I only wanted to highlight their characteristics in this paper. 

13. Subsequent tables 2 and 3 only analyze data for the intervention arm. What was the need for the control. AEs outcome should be compared between the two arms. Where is analyses presented? This analysis that includes the clinical outcomes that compared outcomes between the intervention and control was previously published below. However, usability and acceptability only applies to the intervention arm, so only those men are included. 

Feldacker, C., et al. (2020). "Reducing Provider Workload While Preserving Patient Safety: A Randomized Control Trial Using 2-Way Texting for Postoperative Follow-up in Zimbabwe's Voluntary Medical Male Circumcision Program." JAIDS Journal of Acquired Immune Deficiency Syndromes 83(1): 16-23.

14. Interpretation of ORs appears problematic. When Cis cross 1 results are not statistically significant. This does not appear to be applied in these analyses.. The Table 4 was confusing. The Total SMS responses is a linear regression model. Those CIs do not include “0”. The Potential AE text was a logistic regression model. Therefore, columns 2 and 3 are coefficients and (as noted in the column header) while columns 4 and 5 are ORs, as noted in the column header. Those mostly included “1” and were not significant. We added labels for linear and logistics regression models for clarity. 

6. PLOS authors have the option to publish the peer review history of their article (what does this mean?). If published, this will include your full peer review and any attached files.

Do you want your identity to be public for this peer review? For information about this choice, including consent withdrawal, please see our Privacy Policy.

Reviewer #1: Yes: Limakatso Lebina

Reviewer #2: No

Reviewer #3: No

---

## [Decision Letter · Decision Letter 1]

1 May 2020

Usability and acceptability of a two-way texting intervention for post-operative follow-up for voluntary medical male circumcision in Zimbabwe

PONE-D-19-27561R1

Dear Dr. Feldacker,

We are pleased to inform you that your manuscript has been judged scientifically suitable for publication and will be formally accepted for publication once it complies with all outstanding technical requirements.

With kind regards,

Professor Kwasi Torpey, MD PhD MPH

Academic Editor

PLOS ONE

Additional Editor Comments (optional):

Reviewers' comments:

Reviewer's Responses to Questions

**Comments to the Author**

1. If the authors have adequately addressed your comments raised in a previous round of review and you feel that this manuscript is now acceptable for publication, you may indicate that here to bypass the “Comments to the Author” section, enter your conflict of interest statement in the “Confidential to Editor” section, and submit your "Accept" recommendation.

Reviewer #1: All comments have been addressed

Reviewer #3: All comments have been addressed

2. Is the manuscript technically sound, and do the data support the conclusions?

Reviewer #1: Yes

Reviewer #3: Yes

3. Has the statistical analysis been performed appropriately and rigorously? 

Reviewer #1: Yes

Reviewer #3: Yes

4. Have the authors made all data underlying the findings in their manuscript fully available?

Reviewer #1: Yes

Reviewer #3: Yes

5. Is the manuscript presented in an intelligible fashion and written in standard English?

Reviewer #1: Yes

Reviewer #3: Yes

6. Review Comments to the Author

Reviewer #1: (No Response)

Reviewer #3: (No Response)

7. PLOS authors have the option to publish the peer review history of their article (what does this mean?). If published, this will include your full peer review and any attached files.

Reviewer #1: Yes: Limakatso Lebina

Reviewer #3: No

---

## [Editor Report · Acceptance letter]

5 Jun 2020

PONE-D-19-27561R1 

Usability and acceptability of a two-way texting intervention for post-operative follow-up for voluntary medical male circumcision in Zimbabwe 

Dear Dr. Feldacker:

I'm pleased to inform you that your manuscript has been deemed suitable for publication in PLOS ONE. Congratulations! Your manuscript is now with our production department. 

Kind regards, 

on behalf of

Professor Kwasi Torpey 

Academic Editor

PLOS ONE